# The Effects of Agave Fructans in a Functional Food Consumed by Patients with Irritable Bowel Syndrome with Constipation: A Randomized, Double-Blind, Placebo-Controlled Trial

**DOI:** 10.3390/nu15163526

**Published:** 2023-08-10

**Authors:** Brenda Hildeliza Camacho-Díaz, Martha Lucía Arenas-Ocampo, Perla Osorio-Díaz, Antonio Ruperto Jiménez-Aparicio, Guadalupe Monserrat Alvarado-Jasso, Edén Valfré Saavedra-Briones, Miguel Ángel Valdovinos-Díaz, Elisa Gómez-Reyes

**Affiliations:** 1Centro de Desarrollo de Productos Bióticos, Instituto Politécnico Nacional, San Isidro, Yautepec 62731, Morelos, Mexico; mlarenas@ipn.mx (M.L.A.-O.); posorio@ipn.mx (P.O.-D.); aaparici@ipn.mx (A.R.J.-A.); 2Centro Nacional de Tecnología y Seguridad Alimentaria (CNTA), Crta-Na134 km 53, 31570 San Adrián, Navarra, Spain; gpemonserratjasso@gmail.com; 3Facultad de Nutrición, Universidad Autónoma del Estado de Morelos, Vista Hermosa, Cuernavaca 62290, Morelos, Mexico; eden.saavedra@uaem.edu.mx; 4Laboratorio de Motilidad Gastrointestinal del Departamento de Gastroenterología, Instituto Nacional de Ciencias Médicas y Nutrición Salvador Zubirán, Tlalpan, Ciudad de México 14080, Mexico; miguelvaldovinosd@gmail.com; 5Escuela de Medicina y Ciencias de la Salud, Tecnologico de Monterrey, Campus Ciudad de México, Calle del Puente 222, Col Ejidos de Huipulco, Tlalpan, Ciudad de México 14380, Mexico

**Keywords:** IBS, constipation, prebiotic jelly, agave fructans, functional foods, prebiotics, fiber intake, intestinal movements

## Abstract

Irritable bowel syndrome displays three different subtypes: constipation (IBS-C), diarrhea (IBS-D), and mixed (IBS-M). Treatment with dietary fiber is used, with consideration given both to the chemical composition of the fiber and to the different subtypes of IBS. The IBS-D subtype is usually treated with a low-FODMAPs diet, whereas the IBS-C subtype suggests prebiotics and probiotics to promote microbiota restoration. The aim of this study was to assess the effects of employing agave fructans as the soluble fiber of a jelly (Gelyfun^®^gastro) containing 8 g per serving in the IBS-C group (n = 50), using a randomized, double-blind, time-limited trial for four weeks. We evaluated changes in the frequency and types of bowel movements through the Bristol scale, and the improvement of the condition was evaluated using quality of life (IBS-QOL) and anxiety–depression (HADS) scales. The main results were that the number of bowel movements increased by more than 80%, with at least one stool per day from fifteen days onwards, without a laxative effect for the group treated. Finally, the quality of life with the prebiotic jelly was significantly improved compared to the placebo in all specific domains, in addition to significantly reducing anxiety and depression.

## 1. Introduction

Irritable bowel syndrome (IBS), also known as irritable or spasmodic colon, is a prevalent condition frequently encountered in gastroenterology clinics. It is a chronic functional disorder of the gastrointestinal tract with an unknown underlying cause. The hallmark symptoms of IBS include recurrent abdominal pain or discomfort as well as alterations in bowel function such as changes in frequency or appearance, which can manifest as constipation, diarrhea, or both [1]. According to Roma III criteria, constipation-predominant IBS was identified in patients who had 25% of their stool classified as types 1 or 2 according to the Bristol stool scale, while diarrhea-predominant IBS was determined in patients with 25% of their stool categorized as Bristol stool form types 6 or 7 [2]. However, it is important to note that individual symptoms such as abdominal pain, frequent stools associated with pain, incomplete evacuation, mucus per rectum, abdominal distension, or proctalgia fugax have limited sensitivity and specificity when it comes to diagnosing IBS [3].

Intestinal dysbiosis, characterized by changes in the composition of commensal bacteria in the large intestine, is considered one of the potential factors contributing to the development of IBS. Among the various mechanisms, malabsorption of fructose has been implicated in triggering IBS symptoms. However, it remains unclear why fructose malabsorption occurs and whether this is a consequence of dysbiosis or because of IBS conditions that alter the microbiota–gut–brain axis [3,4].

Similar to global patterns, irritable bowel syndrome (IBS) exhibits a higher prevalence among women in Mexico, irrespective of the subtype or diagnostic criteria [1,3]. IBS significantly contributes to disability, absenteeism from work or school, and increased healthcare expenditures [5]. It is common for patients with IBS to experiment with dietary modifications or restrict the consumption of certain foods even prior to seeking medical attention. The most frequent foods implicated in intestinal discomfort include wheat, corn, dairy products, coffee, tea, and citrus fruits [6]. In certain patients, AU check intended meaning is retained demonstrated improvements in IBS symptoms [7].

Studies have also indicated that manipulating the intake of poorly absorbed short-chain carbohydrates can impact overall gastrointestinal gas production and the composition of gases produced (hydrogen versus methane) in healthy individuals, as well as hydrogen production in patients with IBS [7,8]. Consequently, the treatment of IBS has focused primarily on symptom amelioration with drugs rather than curative interventions for the disease. 

In the conventional management of constipation-predominant IBS, the recommended first-line strategy is to increase fiber intake. This approach is widely advocated in both primary and secondary care settings, with the objective of regulating defecation and addressing delayed intestinal transit [9]. However, there are concerns regarding the potential exacerbation of symptoms in certain individuals when insoluble fiber is consumed.

On the other hand, the inclusion of both fermentable and nonfermentable fibers may offer potential benefits for these patients [1,9,10]. Nonetheless, further information is required to better understand the therapeutic application of fermentable fibers as nutraceutical and food components, particularly regarding the optimal dosage and timing for their incorporation into a daily diet.

Prebiotics are defined as “substrates that are selectively utilized by host microorganisms, conferring a health benefit to the host”. The prebiotics commonly consist of dietary carbohydrates, with inulin-type fructans (ITF) (fructose polymers) and galactooligosaccharides (GOS) (galactose polymers) being extensively studied [11,12]. Fructans are categorized based on their structure and fructosyl linkage, such as inulin, levans, graminans, levan neoseries, and graminan neoseries. 

The significance of inulin-type fructans containing linear β (2→1) linkages in promoting human and bowel health has been firmly established through both in vitro and in vivo studies [12]. These types of fructans have consistently shown associations with increased populations of bifidobacteria and lactobacilli, as well as the production of desirable fermentation end products [12,13]. Agave fructans’ prebiotic activity was studied previously by Velazquez-Martinez et al. [14] as the prebiotic with the best results for the *Lactobacillus paracasei* subsp. Paracasei and *Bifidobacterium bifidum* ATCC 29521 probiotic strains.

The fermentation rate and extent of fructans are influenced by the degree of polymerization. While several studies have investigated linear-chain fructans, limited data are available on branched-chain fructans [12]. Subsequently, Alvarado-Jasso et al. [15] investigated the properties of the agave fructans in a mouse model of obesity. The mice were supplemented with more and less fermentable prebiotics for a duration of six weeks, and various parameters such as body weight gains, levels of short-chain fatty acids, and blood pressure were assessed, along with an increase in fecal excretion [15]. However, it remains uncertain whether the efficacy of prebiotic therapies, such as agave fructans (agavins), has an impact on the short-term progression of IBS gastrointestinal disorders.

Following an extensive analysis of 2332 records, Wilson et al. [11] reported no significant differences between the groups treated with prebiotics or placebos. Furthermore, no variations were observed among the evaluated studies regarding the severity of abdominal pain, bloating, and flatulence. However, notable differences were identified in terms of the increase in the bifidobacterial group and the improvement in or exacerbation of symptoms depending on the specific prebiotic used. It is worth noting that Wilson’s meta-analysis consisted of only one study conducted exclusively on individuals with constipation-predominant IBS, without a direct comparison between the placebo and prebiotic groups concerning stool frequency, which is considered a significant factor. According to the results, anxiety and depressive symptoms were not more prevalent in females than in males. Generally, females are more prone to worse mental health than males, and the treatment of the psychological aspects of the disease can improve the QOL of patients with IBS, emphasizing the importance of the psychological aspects of the condition [16].

Therefore, the aim of this study was to assess the short-term effect of functional food with agave fructans supplementation on the clinical symptoms, quality of life, anxiety, depression, and stool frequency in patients with constipation-predominant IBS.

## 2. Materials and Methods

### 2.1. Materials

The agave fructans used in this study as a source of prebiotic dietary fiber were obtained through a patented process (Mx/a/2015/016512) at CEPROBI, Polytechnic National Institute. It involved extracting a purified powder from *Agave angustifolia* HAW, sourced from the State of Morelos, with a purity of 95% and a degree of polymerization (DP) ranging from 3 to 11, as determined through the mass spectrometer MALDI-TOF MS technique using a microflex^®^ (BRUKER; Billerica, MA, USA), with a 637 nm nitrogen laser and a positive reflector. The samples were dissolved in water (1 mg mL^−1^) and incorporated at a ratio of 1 to 10 in the matrix solution (2,5-dihydroxybenzoic acid saturated in water acetonitrile at a ratio of 70:30). The agave fructans powder was used as an ingredient for the preparation of the hydrated jelly product (Mx/a/2013/013789 patent), with fiber (Gelyfun^®^gastro, Morelos, Mexico) administered to both the placebo and test jelly (130 g per serving). These agave fructans had 3–60 DP, with a majority percentage of 3–11 (Appendix A), according to Velazquez-Martinez et al. [14].

The functional food provided for this study consisted of a prebiotic jelly (Gelyfun^®^gastro) containing agave fructans, available in three flavors: lemon, strawberry, and pineapple. There was a standardized portion size of 130 g per day for each individual serving. The prebiotic jelly used in the test group contained a precise amount of 7.8 g of agave fructans per serving. Both the test and placebo products were packaged and labeled in an analogous manner, ensuring the blinding of the study. 

These products were only distinguishable by the designated ID codes assigned by the “Alimentos BEA” company, which were undisclosed to the doctors and patients involved in the study. Detailed nutritional information regarding the prebiotic jelly, including the jellies given to the placebo and test groups, is presented in Table 1. The determination of protein in foods was performed according to NMX-F-608-NORMEX-2011, the determination of ethereal extract in foods (Soxhlet method) was performed according to NMX-F-615-NORMEX-2018, and the determination of dietary fiber was performed according to NOM-086-SSA1-1994. The Official Methods of Analysis of the Association of Official Analytical Chemists were used in all investigations.

### 2.2. Subjects

The present study was a prospective, double-masked, placebo-controlled trial, as illustrated in Figure 1. Patient recruitment took place at the outpatient clinics of the Department of Gastroenterology and among the working staff of the National Institute of Medical Sciences and Nutrition Salvador Zubirán. Screening of patients was based on the Rome III diagnostic criteria for irritable bowel syndrome (IBS). 

### 2.3. Study Design

After recruitment of patients diagnosed with IBS with a prevalence of constipation, we performed the random assignment of two study groups, namely, an experimental group and a control group. Participants in the experimental group were supplemented with a functional prebiotic jelly (Gelyfun^®^gastro) with agave fructans. The evaluations were carried out 2 weeks and 4 weeks after the beginning of the treatment in an outpatient clinic through questionnaires and physical examination, observing the change in symptoms (Figure 1).

Specifically, individuals with constipation-predominant symptoms were targeted for recruitment, in accordance with predefined inclusion and exclusion criteria and their voluntary willingness to participate. All patients provided informed consent prior to their inclusion in the study. Ethical approval for the study was obtained from the local Research Ethics Committee, and the study was conducted in compliance with the principles set forth in the Declaration of Helsinki (GAS-782-13/14-1) and evaluated and authorized by the INCMNSZ ethics committee. The level of physical activity was evaluated by measuring METS (1296 ± 906) with the instrument validated by the WHO, and energy consumption by means of 24 h reminders, which were analyzed using NutriKcal^®^ VO software. The percentage of carbohydrates, fat, and proteins and the level of physical activity were similar for both groups at the beginning of the study.

### 2.4. Quality of Life (IBS-QOL)

According to Patrick et al. [17] and Drossman et al. [18], a test with 34 items measured on a 5-point Likert scale with eight subscales of dysphoria, interference with activity, body image, health worry, food avoidance, social reactions, sexual issues, and relationships was applied. The obtained scores range from 0 to 100, and the test takes 10 min to complete. Obtaining higher scores in this instrument indicates lower QOL.

### 2.5. Anxiety and Depression (HADS)

The Hospital Anxiety and Depression scale contains 14 items and consists of two subscales: anxiety and depression. Each item is rated on a four-point scale, giving a maximum score of 21 for the total HADS score and anxiety and depression subscales according to Sun Cho et al. [19], Melchior et al. [20], and Groeger et al. [21] for IBS subjects with values ≥ 8. 

### 2.6. Statistical Analysis

The statistical analysis was performed with IBM SPSS statistics V.20 software. A double-blind clinical trial was performed using intention-to-treat analysis on all patients who underwent randomization (n = 50). The additional analysis included only those subjects who completed the study (n = 39). Analysis of anthropometric results revealed no statistically significant differences between these two groups, and the results for the intention-to-treat analysis are presented with Cochran and Wilcoxon comparative analysis for all nonparametric variables. The primary endpoints were changes in constipation, the total QOL, the HADS and its subscales, and the Bristol scale. Comparisons between the treatment and placebo groups were performed using Student’s *t*-test, parametrical analysis was performed using Chi-square tests, and changes during the follow-up were examined using the mixed model analysis of variance (ANOVA) considering significant differences *p* ≤ 0.05 and represented in an APA format in each case.

## 3. Results

### 3.1. Subject Characteristics and Anthropometric Measures

In this pilot exploratory clinical study assessing the efficacy, safety, and tolerability of agave fructans, among the 64 patients who underwent screening for this study, 50 patients fulfilled the inclusion criteria, and these individuals were randomized. Exclusion criteria included a change in treatment for IBS during the clinical trial, displayed by 14 patients (28%); the use of laxatives, displayed by 2 patients (4%); a diet change in the last 4 weeks, displayed by 4 patients (8%); the presence of diarrhea, displayed by 3 patients (6%); and having mixed or undefined IBS, as was the case for 1 patient. Overall, 85% of the subjects included belonged to the female sex, with an average age of 50 ± 2 years. At the beginning of the study, the body composition of the subjects was evaluated using electrical impedance, displaying an average fat content of 37 ± 9%, an average body mass index of 28.16 ± 4 kg/m^2^, and an average waist circumference of 90.6 ± 12.3 cm. Patients were questioned about their quality of life and the presence of stress, and the most frequently reported symptoms were bloating in 89% of cases, the presence of gas in 52% of cases, gastritis and reflux in 58% of cases, abdominal pain in 50% of cases, and the presence of stress in 91.6% of cases. 

Table 2 shows that there was no weight loss due to the consumption of agave fructans using jelly as a vehicle, since there were no significant differences when comparing the treatment group with the placebo group, and the different body measures were also similar. During the treatment, none of the groups modified their calorie intake or significantly modified their consumption of nutrients, except for an increase in fiber consumption in the study group, who consumed up to 23 g of fiber per individual during the treatment. The sum of the fiber consumed in the diet was accounted for in addition to the consumption of the jelly with fiber. 

Regarding the body measurements of both groups, they were similar, which demonstrates the homogeneity of the samples. The energy consumed by the diet for both groups was the same (*p* = 0.218), while the metabolic activity was different between groups (*p* = 0.005), being higher for the intervention group. Meanwhile, fiber intake increased during the prebiotic jelly intervention (16.09 g to 23.48 g) after 30 days of treatment compared to the placebo group (13.98 g to 10.66 g).

### 3.2. Gastrointestinal Symptoms

Both groups evaluated responded significantly to the protocol; however, the group that consumed agave fructans using jelly as a vehicle showed a constipation decrease of more than 83% in the population evaluated, in contrast to 45% of the placebo group, which responded from 15 days of intervention in both cases. It is important to mention that this percentage in the case of patients who consumed jelly with agave fructans was modified to only a 67% improvement in constipation, compared to the 56% improvement in the placebo group. Thus, there was a higher number of individuals who responded to treatment with fructans at 15 days than at 30 days (*p* = 0.014).

Table 3 presents the demographic data of 50 patients belonging to either the study group or the placebo group. In general, the patients were mostly women; 100% of the patients in both groups presented with IBS with a predominance of constipation and colitis as part of the symptoms for both groups; and more than 50% of patients in both groups also presented with gastritis and reflux, while a smaller percentage of the population (5–20%) presented with nausea and vomiting symptoms [4].

At the start of treatment, patients in both groups (placebo and agave fructans treatment) presented with additional undesirable symptoms characteristic of IBS, like gastritis, reflux, nausea, and vomiting symptoms, as reported by Konturek et al. [22]. After thirty days of Gelyfun^®^gastro prebiotic jelly treatment, most of these symptoms were reduced. However, gastritis was reduced and flatulence increased significantly between the placebo and study group. Despite flatulence increasing, for abdominal pain, there were no significant changes with Gelyfun^®^gastro prebiotic jelly consumption.

Furthermore, Table 3 shows that out of the 50 patients evaluated, 28% were excluded because they adopted another treatment for the control of constipation or the improvement of symptoms due to IBS with a predominance of constipation during the protocol, and another 6% of the population were excluded due to diarrhea at the beginning of the protocol, since they could have been considered as having mixed-type IBS [23]. It should be noted from Table 4 that the only symptoms in which there was a significant difference (0.014 to 0.037) between the study group and the placebo group were constipation, flatulence, and nausea, which is consistent with those described by Niv et al. [4].

### 3.3. Stool Characteristics and Bowel Movements

In this study, we evaluated the treatment’s impact not only on the decrease or increase in characteristic symptoms (Table 3), such as constipation, flatulence, and bloating, but also on the changes in the frequency and consistency of bowel movements, using the Bristol scale as a reference. Then, we subclassified the types of bowel movement into constipation, normal, and diarrhea (Table 4).

For the construction of Table 4, it was necessary to subclassify stool types through the Bristol scale (scale 1–7) according to gastrointestinal conditions, considering patients with a Bristol scale value of 1 or 2 as having constipation; patients with a Bristol scale value of 3, 4, or 5 as normal; and finally, patients with a Bristol scale value of 6 or 7 as having diarrhea, as established by Ersryd et al. [24] and Blake et al. [25], who used the Bristol scale as a tool to identify different subtypes of IBS, comparing patients diagnosed using the Rome II and Rome III criteria.

In Table 4, where the groups are compared over a period of 15 and 30 days of treatment and the improvement in their gastrointestinal condition as measured by the Bristol scale is compared between the groups, it can be observed that there was not only an improvement in the gastrointestinal condition during the protocol but also an increase in the number of bowel movements per week. This was observed for individuals receiving the treatment, while individuals receiving the placebo experienced diarrhea and an increase in the number of bowel movements per week to as high as 18.

### 3.4. Quality of Life (IBS-QOL) and Subscales

In this study, the use of 8 g of agave fructans as an ingredient of the functional food Gelyfun^®^gastro increased the quality of life from 61.47% to 76.06%, *p* = 0.001, in a period of 30 days, in comparison to the placebo, which increased it from 70.76% to 79.94%, *p* = 0.003. According to Drossman et al. [18], an improvement in the IBS-QOL overall score of ≥14 points from the baseline to a time point of interest could be considered a minimal clinically important difference between the treatment and placebo. From this analysis, regarding treatment satisfaction, the mean change in the overall and subscale IBS-QOL scores with the Gelyfun^®^gastro, majorly, was ≥14 (14.00–16.00), except for the interference with activity subscale (13.86), and for the placebo, satisfaction was ≤12 for all IBS-QOL subscales. These results overall show a better response in the treatment group compared to the placebo group, even though significant differences were not observed between the groups.

Regarding the study of quality of life evaluated through the IBS-QOL questionnaire (Table 5), in general, both groups experienced improvements in their quality of life in every one of the subcategories assessed; however, there was no response with greater or less significance between the two groups evaluated. Additionally, the overall scores for IBS-QOL for subtype IBS-C and for all subcategories were similar to those reported by Cho et al. [19].

### 3.5. Anxiety and Depression (HADS) Score

Commonly, the mean score for depression in IBS patients is higher than that in healthy patients, as is that for anxiety (≥8) [19]. In this study, IBS-C patients had severe anxiety and depression scores, and there were no statistical differences in either anxiety or depression scores between the placebo group and the diet-supplemented group.

The overall baseline anxiety and depression scores were observed in IBS-C patients, being 29.04 for the Gelyfun^®^gastro group and 21.58 for the placebo group. The mean baseline HADS scores for anxiety and depression in IBS patients were 13.92 and 12.77 for anxiety and 15.13 and 14.58 for depression for the treatment and placebo groups, respectively. Both anxiety and depression baseline scores corresponded to the scores for a severe IBS condition reported by Banerjee et al. [26] for different IBS subtypes and severity levels.

After treatment, there were no significant differences in depression scores, but when adjusting for baseline differences, a greater improvement in anxiety was found in the Gelyfun^®^gastro group with respect to the placebo group. Pinto-Sanchez et al. [27] identified decreased depression compared to the baseline score after 6 weeks of treatment with probiotics (range 1.16–3.38). In the present study, the response was more prominent with the functional prebiotic jelly, with a mean change from the baseline of 3.55–3.92, compared with 2.69–4.12 for the placebo group.

## 4. Discussion

There is still no standard for the treatment of IBS, which means that when new therapies are tested, they are usually compared to the placebo. However, placebo response rates in this disease are high: 30% to 40% of patients experience relief or resolution of symptoms, as shown by Ford et al. [28]. In this study, the placebo group displayed a 44% rate of response, compared to a rate of 83% for constipation improvement in the first 15 days of treatment.

Agave fructans, as an ingredient in functional foods in the diet, also serve as a support for the gut microbiota, which in turn favorably influences the production of fermentation products with beneficial metabolic effects, such as short-chain fatty acids [29]. In a previous study, Alvarado-Jasso et al. [15], using agave fructans (Mx/a/2013/013789 patent) in obese mice, found that SCFA butyrate concentrations and loss in weight were significantly higher in obese mice supplemented with agave fructans than in mice which were not supplemented with agave fructans. 

In the present study, we did not observe significant weight changes caused by the jelly functional food treatment. This is in contrast to what was reported by Silva-Adame et al. [30], who mentioned that after the administration of 10 g of agave fructans per day, a reduction in weight from baseline conditions could be observed, and in this study, only 7.8 g of agave fructans was administered in a portion of Gelyfun^®^gastro. In general, the anthropometric data showed changes after thirty days of treatment in the study group. 

Many clinical trials have evaluated how a low-FODMAP diet (LFD) affects IBS symptoms. These studies indicate that a low-FODMAP diet helps to relieve symptoms. However, fewer protocols have been evaluated, suggesting that a low-FODMAP diet could affect the microbiota and lead to a state of dysbiosis. Therefore, the low-FODMAP diet remains a controversial topic. According to Martínez Vázquez [31], in three different meta-analyses [32,33,34], low-FODMAP diets were found to have special indications in each IBS subtype. For example, in IBS-D with a predominance of abdominal pain, a major positive effect can be observed, whereas in IBS-C, limited effects can be seen. 

Additionally, it is well-known that microbial changes can worsen intestinal symptoms associated with IBS-C, such as visceral pain, low-grade inflammation, and changes in stool frequency. Substantial evidence indicates that the microbiota is one of the primary factors affecting IBS in certain patients. This process is unclear but may be due to the transient alteration of the microbiota composition postinfection and ongoing dysbiosis in the presence of low-grade mucosal inflammation [35,36]. Nevertheless, there is little information that clarifies whether there is a specific role for prebiotics in the IBS-C subset of patients with IBS, and in particular whether there is a role for prebiotic carbohydrates that modulate the microbiota without leading to specific symptoms such as abdominal pain, gases, and bloating, or that affect stool output types and frequency.

On the other hand, the fact that most of the symptoms evaluated do not present significant differences, except for those mentioned above (constipation, flatulence, and gastritis), indicates that, contrary to what several authors asserted regarding the effect of the low-FODMAP diet on IBS due to the presence of some undesirable symptoms, the implementation of agave fructans as a prebiotic using the Gelyfun^®^gastro jelly as a functional food (Mx/a/2013/013789), referred to in the patent as “fiber hydrated”, contributed to the easy fermentation of fiber by the microbiota in the colon and consequently the easing of undesirable symptoms. Most of these symptoms demonstrated decreased incidence after 15 to 30 days in patients supplemented with the functional food and not in patients given the placebo (Table 4).

Di Rosa et al. [37] observed in a meta-analysis that IBS patients treated with insoluble fiber may experience an exacerbation of their symptoms and experience minimal relief, while the administration of psyllium as a soluble fiber could be more effective in reducing symptoms. On the other hand, Niv et al. [4] determined that in a group of patients with IBS who displayed a predominance of constipation and were treated with *Lactobacillus reuterie* (probiotic), constipation and flatulence were the only parameters displaying significant differences compared to the placebo group.

In addition, in this protocol, the group treated with agave fructans presented a significant difference over the 30 days of treatment compared to the placebo in terms of initial abdominal pain and stress condition factors which were subsequently related to the evaluation of quality of life. Andrae et al. [38] suggested that the effectiveness of treatments in controlling IBS is closely related to the psychosomatic condition of the disease.

Once the reclassification was performed, it was possible to establish a significant difference between the study group and the placebo group in terms of the effect of fructans as a useful treatment in conditions such as IBS with a predominance of constipation, where constipation is the main factor associated with the symptoms of these patients (abdominal pain and inflammation, pain when defecating, and flatulence with an unpleasant smell). The use of FODMAPs (prebiotics), according to Staudacher et al. [39], is not recommended in the treatment of IBS because their fermentation in the intestine causes undesirable symptoms such as excessive flatulence and abdominal swelling. However, in this study, there was no way to differentiate between patients with IBS with diarrhea, mixed IBS, and IBS with constipation; instead, a general recommendation is provided for the reduction in FODMAPs in the diet in patients diagnosed with IBS.

Therefore, it is possible to consider that the improvement in the symptoms associated with IBS-C was due to (a) the type of prebiotic used (Mx/a/2015/016512); (b) the supplemented dose between 5 and 15 g; (c) the duration of the treatment of at least 15 days; and (d) the method of administration of the agave fructans, which was through the Gelyfun^®^gastro functional food.

Regarding the type of prebiotic, according to McRorie, [40] soluble, β-glucan, and nonviscous fermentable fibers (e.g., inulin or fructooligosaccharides) do not provide laxative effects in comparison with psyllium fiber, which is considered as a medicinal fiber due to its ability to absorb water and form gel and its mechanism of action, according to Remes-Troche et al. [41]. 

Linear and branched fructans like agave fructans can normalize gut functionality by producing products of fermentation such as SCFA in specific high concentrations of butyrate [42]. Hence, this causes a significant increase in the number of bowel movements and the normalization of stool type. Unlike the case of chicory inulin (the most evaluated prebiotic) in the studies of Glibowski et al. [43] and Micka et al. [44], no significant differences in stool consistency or constipation-associated symptoms compared with the placebo were found.

According to the dose and duration of treatment, it is important to highlight that according to McRorie [40], in protocols with inulin, there were no significant changes compared to treatments with chicory inulin in terms of the improvement of constipation in general, using doses of 5 to 20 g/day in a treatment period of 1 to 4 weeks. This is contrary to what was obtained with agave fructans in this work, where with only 8 g, after the first 15 days of treatment with Gelyfun^®^gastro, in 70–76% of the test group, 12 bowel movements per week were obtained with a normal consistency (types 3, 4, and 5) according to the Bristol scale, displaying a significant difference with regard to the beginning of the treatment and the placebo group. 

On the other hand, Silva-Adame et al. [30], in relation to the dosage used and the duration of treatment for agave fructans, observed the same tolerability for 10–12 g per day in normal and obese patients, without any significant differences in terms of symptoms related to tolerability (satiety, appetite, metabolic markers, and body composition). However, in this study, there were no changes in weight, but there was improvement in the tolerability of certain foods in the normal diet of patients, considering that in the present study, no change in weight was included within the criteria. The daily diet of each of the patients was only monitored through daily diet reminder instruments. This finding was relevant given that there were no restrictions on the diets of the patients during the intervention, as occurs in alternative treatments, such as the low-FODMAP diet [8,12,36,39].

Finally, regarding the method of administration, this study used a functional food, i.e., a functional hydrated fiber (Gelyfun^®^gastro), to administer agave fructans, unlike the majority of clinical protocols in humans, in which agave fructans were administered in the form of powder, capsules, or tablets, in which no significant differences in the improvement of IBS-C symptoms have been evidenced [37], even for people only with constipation [40]. In the particular case of Glibowski et al. [43], processed apple beverages supplemented with inulin were evaluated in constipated patients and an the increase in stool frequency was observed, reaching 12 times/week, without showing the consistency and shape of the stool according to the Bristol scale, as was evaluated and interpreted in this study in Table 5.

The effects of functional foods with fructans on improving symptoms and quality of life in IBS-C have been studied previously in humans. The present study is the first clinical trial to date that uses agave fructans as an ingredient in functional food for IBS-C treatment.

Previous clinical studies of constipation and IBS-C have used functional foods as modes of treatment, for example, dairy products [42], juices [43], and gels [45,46], using fructans such as inulin of chicory alone or symbiotically. However, most of these studies only tested the effectiveness of this type of soluble fiber as an ingredient in concentrations of less than 5 g per daily serving, most likely due to the increase in the degree of viscosity that can occur in food when adding these fiber concentrations [47]. Therefore, the addition of agave fructans that present a higher degree of solubility compared to inulin allowed us to add a more significant amount of soluble fiber without significantly modifying the characteristics of the functional prebiotic gelatin. 

For this reason, this study suggests that the definition of the term “functional” can differ from that provided by the Panel on the Definition of Dietary Fiber Staff from the Institute of Medicine [48], which states that “functional fiber consists of isolated, nondigestible carbohydrates that have beneficial physiological effects in humans”. This research additionally postulates that functional prebiotic fiber should be consumed in conditions similar to those found naturally in food (hydrated fibers) in order to be named “functional”. In this sense, using less than 5 g of prebiotic fiber provides an increase in dietary fiber in the diet that does not have a physiological effect, while using more than 15 g can have a laxative effect due to excess or nontolerability to this type of fiber. Using the correct amount has advantages for obtaining SCFA of the type and amounts that are conducive to regularizing intestinal function in a continuous minimum period of 15 days.

On the other hand, when using this type of treatment, in most cases, improvements in the frequency and consistency of bowel movements have been obtained, and in very few cases, the possibility of significantly improving the symptoms of the condition has been seen, as well as a limited or insignificant improvement in quality of life, as well as nonsignificant changes in the anxiety and depression scores evaluated using the HADS instrument.

The results of the present investigation showed significant improvements in the quality of life of IBS-C patients using a functional prebiotic jelly as a continuous 4-week treatment. These results differ from those in a recent meta-analysis by Willson Rossi et al. [11], which found that there was no significant effect of prebiotics such as inulin on IBS-QOL in three previous studies that used the validated IBS-QoL questionnaire, and those in the study by Roberts et al. [49], which performed an evaluation of probiotics in IBS, where they found positive changes in the IBS-QOL total score until 8 weeks. 

Generally, females are more prone to worse mental health than males, and the effective treatment of the psychological aspects of the disease including anxiety and depression can improve the QOL of patients with IBS. So, the treatment of a functional prebiotic fiber (Gelyfun^®^gastro) could reduce visceral hypersensitivity with SCFA, attenuate anxiety behaviors, and modulate brain activity in IBS-C patients.

According to Freijy et al. [50], ingesting prebiotics provided better results than ingesting probiotics or symbiotics, with which there was no evidence of symptom improvement for anxiety and depression. They reported positive changes in anxiety at a 5 g daily dose of prebiotics, the minimum level at which a diet supplemented with prebiotics conferred psychological benefits. 

## 5. Conclusions

This study is the first report that assesses the short-term effects of agave fructans administration on the clinical symptoms of IBS-C. Through the ingestion of almost eight grams of agave fructans in a functional food (prebiotic jelly) for a period of 15 days, the frequency of fecal evacuation increased to 12 per week, the Bristol scale was regularized to type 4, and patients’ anxiety and depression levels were modified with respect to the baseline values in both evaluated groups. Finally, it was possible to improve the quality of life of patients with IBS of the constipation subtype, with significant effects on the food avoidance, social reaction, and sexual domains with respect to the placebo group with a low amount of 7.8 g of agave fructans, without a laxative effect or nontolerability. However, the evaluation of the intestinal microbiota and a possible description of the mechanism of action of the tested product are necessary.

## Figures and Tables

**Figure 1 nutrients-15-03526-f001:**
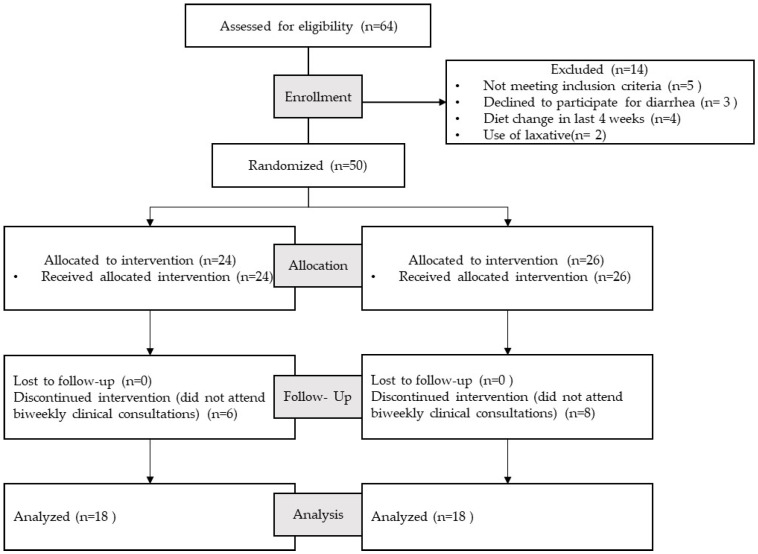
Diagram of study design.

**Table 1 nutrients-15-03526-t001:** Gelyfun^®^gastro and placebo jelly nutrition facts.

Gelyfun^®^gastro	Test Jelly		Placebo Jelly	
Nutrition Facts	g/Serving	kcal/Serving	g/Serving	kcal/Serving
Carbohydrates	13.2	52.8	13.2	52.8
Proteins	2.9	11.6	2.9	11.6
Lipids	0.0	0.0	0.0	0.0
Fiber (as powder with 95% agave fructans)	7.8	11.7	___	___
	Total	76.1	Total	64.4

**Table 2 nutrients-15-03526-t002:** Anthropometric data for the baseline and within-group comparisons after 30 days of the intervention.

Study Parameter	Baseline Study Group (n = 24)	30 Days Baseline Study Group (n = 18)	*p* Value * (Change from Baseline)	Baseline Placebo Group (n = 26)	30 Days Placebo Group (n = 18)	*p* Value * (Change from Baseline)	*p* Value ** (Study vs. Placebo Group 30 Days)
Weight (kg)	70.98	71.38	0.846	67.99	58.87	0.951	0.373
BMI (kg/m^2^)	28.34	28.11	0.765	27.65	24.15	0.839	0.494
Waist circumference (cm)	90.13	91.03	0.607	89.43	77.75	0.607	0.710
MUAC(cm)	31.16	31.18	0.684	31.44	27.43	0.072	0.039 **
Body fat (%)	38.47	38.42	0.181	37.47	32.80	0.090	0.571
Fat-free mass (%)	60.99	62.04	0.370	62.52	53.78	0.102	0.585
Carbohydrates (%)	53,78	53.70	0.957	55,45	51.71	0.344	0.319
Proteins (%)	16.74	18.55	0.145	16.77	17.50	0.314	0.594
Lipids (%)	31.0	29.65	0.812	29.62	31.97	0.693	0.207
Fiber (g)	16.09	23.48	0.021 *	13.98	10.66	0.171	0.388
Energy (kcal)	1771.2	1841	0.625	1614	1580.86	0.850	0.218
MET (kcal/kg/h)	1687	1315	0.189	941.9	568	0.310	0.005 **

BMI: body mass index, SD: standard deviation, MUAC: mid-upper arm circumference, MET: metabolic equivalent of task. ** Study vs. placebo group after 30 days; * change from baseline.

**Table 3 nutrients-15-03526-t003:** Symptom scores, differences between baseline, and within-group comparisons at 15 and 30 days.

Study Parameter	Baseline Study Group n = 24 (n, %)		Day 15 Study Group n = 18 (n, %)		Day 30 Study Group n = 18 (n, %)		*p* Value * (Change from Baseline)	Baseline Placebo Group n = 26 (n, %)		Day 15 Placebo Group n = 18 (n, %)		Day 30 Placebo Group n = 18 (n, %)		*p* Value * (Change from Baseline)	*p* Value ** (Study Group/ Placebo Group 30 Day)
Diarrhea	1	4.2%	3	16.7%	2	11.1%	0.549	2	7.7%	3	16.7%	2	11.1%	0.174	0.694
Constipation	24	100%	3	16.7%	6	33.3%	0.000 *	26	100%	10	55.6%	8	44.4%	0.000 *	0.014 **
Gastritis	11	45.8%	2	11.1%	5	27.8%	0.016 *	18	69.2%	8	44.4%	9	50.0%	0.018 *	0.217
Reflux	14	58.3%	4	22.2%	4	22.2%	0.125	15	57.7%	8	44.4%	8	44.4%	0.368	0.570
Vomiting	2	8.3%	0	0.0%	0	0.0%	0.135	3	11.5%	0	0.0%	0	0.0%	0.135	0.817
Bloating	11	45.8%	14	77.8%	11	61.1%	0.368	23	88.5%	15	83.3%	13	72.2%	0.016 *	0.161
Flatulence	8	33.3%	15	83.3%	13	72.2%	0.368	19	73.1%	16	88.9%	11	61.1%	0.074	0.000 **
Abdominal pain	7	29.2%	6	33.3%	3	16.7%	0.097	19	73.1%	13	72.2%	9	50.0%	0.093	0.702

** Study vs. placebo group after 30 days; * change from baseline.

**Table 4 nutrients-15-03526-t004:** Gastrointestinal disorder level with subclassification using the Bristol Scale.

	Gastrointestinal Disorder Bristol Subscale	Bowel Movements per Week	% Study Group	Bowel Movements per Week	% Placebo Group
Basal	Constipation	2	69.6% _a_	4	76.9% _a_
	Normal	3	30.4% _a_	3	23.1% _a_
	Diarrhea	0	0.0% _a_	0	0.0% _a_
Day 15	Constipation	7	10.5% _b_	8	41.7% _a_
	Normal	12	73.7% _b_	9	54.2% _a_
	Diarrhea	14	15.8% _a_	14	4.2% _a_
Day 30	Constipation	4	25.0% _a_	4	25.0% _a_
	Normal	12	70.0% _b_	10	55.0% _a_
	Diarrhea	7	5.0% _a_	18	20.0% _a_

_a,b_ Significant differences of *p* ≤ 0.05 between columns.

**Table 5 nutrients-15-03526-t005:** Quality of life score and subscales and HADS score and subscales.

Study Parameter	Baseline Study Group (n = 24) ±(SD)	30 Days Study Group (n = 18) ±(SD)	Mean Change from Baseline	*p* Value *	Baseline Placebo Group (n = 26) ±(SD)	30 Days Placebo Group (n = 18) ±(SD)	Mean Change from Baseline	*p* Value *	*p* Value **
Overall IBS-QOL (%)	61.47 ±5.84	76.06 ±4.24	14.59	0.001 *	70.76 ±4.52	79.94 ±3.45	9.18	0.003 *	0.456
Dysphoria	65.13 ±9.19	79.13 ±8.04	14.00	0.001 *	75.00 ±9.35	84.25 ±7.69	9.25	0.005 *	0.810
Interference with activity	64.00 ±6.58	77.86 ±4.67	13.86	0.004 *	70.57 ±7.56	81.57 ±6.11	11.00	0.002 *	0.723
Body image	55.75 ±5.01	71.75 ±4.15	16.00	0.002 *	64.75 ±5.09	74.50 ±4.81	9.75	0.001 *	0.666
Health worry	46.00 ±2.79	61.33 ±3.23	15.33	0.005 *	57.33 ±2.35	69.33 ±2.90	12.00	0.003 *	0.504
Food avoidance	56.00 ±3.28	71.33 ±2.95	15.33	0.006 *	62.67 ±3.27	69.33 ±2.53	6.67	0.188	0.203
Social reaction	63.50 ±4.15	78.50 ±3.38	15.00	0.001 *	78.67 ±2.76	84.00 ±3.24	5.33	0.368	0.958
Sexual	64.50 ±2.81	79.50 ±2.62	15.00	0.017 *	79.00 ±3.15	86.00 ±2.12	7.00	0.096	0.222
Relationship	69.33 ±3.36	83.33 ±2.56	14.00	0.006 *	74.00 ±1.91	84.00 ±1.64	10.00	0.003 *	0.719
Overall HADS	29.04 ±4.60	21.58 ±2.84	7.46	0.003 *	27.35 ±5.47	20.54 ±3.11	6.81	0.015 *	0.333
Anxiety	13.92 ±1.00	10.00 ±0.75	3.92	0.003 *	12.77 ±3.89	10.08 ±1.92	2.69	0.051	0.215
Depression	15.13 ±0.75	11.58 ±0.16	3.55	0.006 *	14.58 ±0.51	10.46 ±0.87	4.12	0.007 *	0.609

** Study vs. placebo group after 30 days; * change from baseline.

## Data Availability

Not applicable.

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
