# Peer review of "The Effects of Agave Fructans in a Functional Food Consumed by Patients with Irritable Bowel Syndrome with Constipation: A Randomized, Double-Blind, Placebo-Controlled Trial"

_nutrients, 2023, doi:10.3390/nu15163526_

Round 1

Reviewer 1 Report

Thank you for submitting the manuscript "Effect of agave fructans in a functional food consumed by patients with IBS-C: a randomized, double-blind, placebo-controlled trial" to Nutrients. Overall, the study is interesting. I have concerns relating especially to the authors' conflict of interest. In addition, authors need to consider performing a good English language review to improve the quality and clarity of the manuscript. Here are my suggestions for the manuscript.

1) Even though IBS-C is a common abbreviation, it would be interesting if the title did not include any abbreviation to make it clearer and easier to search as an online indexer. Also, titles do not have a full stop.

2) Abstract

Line#20: Consider rewriting "The use of dietary fiber has been used"

Line#24: Something doesn't seem right in " as ingredients of functional food Gelyfun®gastro"

Line#25: Include study duration.

Line#27: include "and" before "anxiety"

Line#30: Up to this point in the abstract it looks like three different products have been tested: agave fructans, functional food Gelyfun®gastro, and prebiotic jelly. Make it clearer the first time it appears in the abstract.

Consider rewriting the abstract as a whole, making the work hypothesis clearer and more objective, what was done to achieve this hypothesis and what was obtained as a result.

3) Introduction

This topic is too long and needs to be shortened. Although everything said in it is important for understanding the research, it can be more objective and summarized.

4) Material and methods

Line#143: if the product is in the process of obtaining a patent, this means that the development of the study has a commercial objective, that is, it needs to be declared in conflict of interest.

Figure S1: if it is a figure from a supplementary file, it does not need to appear in the body of the text of the manuscript. Also, the resolution of the figure needs to be improved.

Line#157: In many countries there is a limiting maximum percentage of inclusion of dietary fiber from fructans in the diet. Is this product meeting this percentage? These fructans are what products? FOS or inulin?

Line#174: will be?

5) Results

It seems obvious to me that there can be improvement if the amount of fiber in the diet is improved. This placebo is more like a negative control as it does not provide any amount of fiber from another source to the group.

6) Discussion

It seems to me that this patent has been cited so many times that I wonder if all conflicts of interest were actually reported by the authors.

6) Discussion

It seems to me that this patent has been cited so many times that I wonder if all conflicts of interest were actually reported by the authors.

At the end of this topic consider including the limitations of the study, for example the evaluation of the intestinal microbiota and a demonstration of the mechanism of action of the tested product.

The authors need to consider performing a good English language review to improve the quality and clarity of the manuscript.

Author Response

Yautepec; Morelos Mexico, 20th July 2023

Dr. Jan Willem Van der Kamp

Guest Editors of Nutrients from MDPI

Special Issue " Dietary Fiber and Digestive Health"

Dear Editor,

Brenda Hildeliza Camacho Díaz, Martha Lucía Arenas Ocampo, Perla Osorio Díaz, Antonio Ruperto Jiménez Aparicio, Guadalupe Monserrat Alvarado Jasso, Edén Valfré Saavedra Briones, Miguel Ángel Valdovinos Díaz, Elisa Gómez Reyes, as exclusive authors of the text “Effect of agave fructans in a functional food consumed by patients with IBS-C: a randomized, double-blind, placebo-controlled trial.” by declare that we have considered each of the observations.

Thank you for the valuable comments provided by Reviewer 1 on the manuscript. We are also grateful for the opportunity to be a part of Nutrients publications with the Manuscript ID nutrients-2524524. We hope that our responses are in agreement with the ones requested by the journal. We have made every effort to respond promptly. We are attaching the list of answers in the following table.

Reviewer 2 Report

The aim of the manuscript entitled “Effect of agave fructans in a functional food consumed by patients with IBS-C: a randomized, double-blind, placebo-controlled trial” is to evaluate the short-term effect of agave fructans on the clinical symptoms, quality of life, anxiety, depression, and stool frequency in patients with IBS constipation predominant.

This objective falls within the objective of Nutrients (MDPI), which is to publish papers that provide novel insights into the effects of nutrition on human health or novel methods for assessing nutritional status. 

This paper proved to be very relevant in its practical implications and scientifically detailed. 

The introduction is well written and perfectly describes irritable bowel syndrome (IBS), its possible causes and the therapeutic approaches used to alleviate its symptoms. This is followed by a reference is mase to prebiotics, with a focus on agave fructans. 

The purpose of the work, as stated in the introduction, is respected, and reinforced by the experimental part and the discussion of the results. 

Overall, the manuscript is of good quality, fluent and clear in content. 

Some suggestions are made which, in my opinion, could further improve the overall quality of the manuscript. 

Introduction

Lines 70-72: I would recommend including a bibliographical reference. 

            Materials and Methods

Line 146: After mentioning the analytical technique (MALDI-TOF MS), I would recommend also including the operating conditions used. 

Lines 159-162: As you have referred to the nutritional information in Table 1, I would recommend that the manuscript also includes the analytical techniques used for these analyses. This addition would, in my opinion, further enhance the quality of the manuscript. 

Conclusion

Lines 493-495: I would recommend combining the two sentences into one sentence.

Author Response

Yautepec; Morelos Mexico, 20th July 2023

Dr. Jan Willem Van der Kamp

Guest Editors of Nutrients from MDPI

Special Issue " Dietary Fiber and Digestive Health"

Dear Editor,

Brenda Hildeliza Camacho Díaz, Martha Lucía Arenas Ocampo, Perla Osorio Díaz, Antonio Ruperto Jiménez Aparicio, Guadalupe Monserrat Alvarado Jasso, Edén Valfré Saavedra Briones, Miguel Ángel Valdovinos Díaz, Elisa Gómez Reyes, as exclusive authors of the text “Effect of agave fructans in a functional food consumed by patients with IBS-C: a randomized, double-blind, placebo-controlled trial.” by declare that we have considered each of the observations.

Thank you for the valuable comments provided by Reviewer 2 on the manuscript. We are also grateful for the opportunity to be a part of Nutrients publications with the Manuscript ID nutrients-2524524. We hope that our responses are in agreement with the ones requested by the journal. We have made every effort to respond promptly. We are attaching the list of answers in the following table.

Round 2

Reviewer 1 Report

Thank you for accepting some of my suggestions for the manuscript.

I still insist that the introduction should be changed (it is too long and it is difficult to understand the justification of the work based on it). Although the authors say they have carried out this suggestion, there are no marked changes to the text.